# Primary Epstein-Barr virus infection with and without infectious mononucleosis

Klaus Rostgaard[1]*, Henry H. Balfour Jr.[2,3], Ruth Jarrett[4], Christian Erikstrup[5], Ole Pedersen[6], Henrik Ullum[7], Lars Peter Nielsen[8], Marianne Voldstedlund[9], Henrik Hjalgrim[1,10]

**1** Department of Epidemiology Research, Statens Serum Institut, Copenhagen, Denmark, **2** Department of Laboratory Medicine and Pathology, University of Minnesota Medical Center, Minneapolis, MN, United States of America, **3** Department of Pediatrics, University of Minnesota Medical Center, Minneapolis, MN, United States of America, **4** MRC—University of Glasgow Centre for Virus Research, University of Glasgow, Glasgow, United Kingdom, **5** Department of Clinical Immunology, Aarhus University Hospital, Aarhus, Denmark, **6** Department of Clinical Immunology, Næstved Hospital, Næstved, Denmark, **7** Department of Clinical Immunology, Copenhagen University Hospital, Copenhagen, Denmark, **8** Danish National Biobank, Statens Serum Institut, Copenhagen, Denmark, **9** Department of Infectious Epidemiology, Statens Serum Institut, Copenhagen, Denmark, **10** Department of Haematology, Copenhagen University Hospital, Copenhagen, Denmark

* klp@ssi.dk

**Data Availability Statement:** All relevant data are within the paper and its Supporting Information files.

## Abstract

### Background

Infectious mononucleosis (IM) is a common adverse presentation of primary infection with Epstein-Barr virus (EBV) in adolescence and later, but is rarely recognized in early childhood where primary EBV infection commonly occurs. It is not known what triggers IM, and also not why IM risk upon primary EBV infection (IM attack rate) seemingly varies between children and adolescents. IM symptoms may be severe and persist for a long time. IM also markedly elevates the risk of Hodgkin lymphoma and multiple sclerosis for unknown reasons. The way IM occurrence depends on age and sex is incompletely described and hard to interpret etiologically, because it depends on three quantities that are not readily observable: the prevalence of EBV-naïve persons, the hazard rate of seroconverting and the attack rate, i.e. the fraction of primary EBV infections that is accompanied by IM. We therefore aimed to provide these quantities indirectly, to obtain epidemiologically interpretable measures of the dynamics of IM occurrence to provide etiological clues.

### Methods and findings

We used joint modeling of EBV prevalence and IM occurrence data to provide detailed sex- and age-specific EBV infection rates and IM attack rates and derivatives thereof for a target population of all Danes age 0–29 years in 2006–2011. We demonstrate for the first time that IM attack rates increase dramatically rather precisely in conjunction to typical ages of puberty onset. The shape of the seroconversion hazard rate for children and teenagers confirmed a priori expectations and underlined the importance of what happens at age 0–2 years. The cumulative risk of IM before age 30 years was 13.3% for males and 22.4% for

**Funding:** The study was supported by Ulla og Mogens Folmer Andersen's Foundation (grant number 100041-1 to HH), Helsefonden (https://helsefonden.dk; grant number 19-B-0352 to KR), and the University of Minnesota Foundation (https://give.umn.edu; grant number 20707 to HB). The funders had no role in the study design, data collection and analysis, decision to publish, or preparation of the manuscript.

**Competing interests:** The authors have declared that no competing interests exist.

**Abbreviations:** EBV, Epstein-Barr virus; EBNA, EBV nuclear antigen; DBDS, Danish Blood Donor Study; IM, infectious mononucleosis; LL, log-likelihood; NPR, National Patient Register; VCA, EBV capsid antigen.

females. IM is likely to become more common through delaying EBV infection in years to come.

## Conclusions

The change in attack rate at typical ages of puberty onset suggests that the immunologic response to EBV drastically changes over a relatively short age-span. We speculate that these changes are an integrated part of normal sexual maturation. Our findings may inform further etiologic research into EBV-related diseases and vaccine design. Our methodology is applicable to the epidemiological study of any infectious agent that establishes a persistent infection in the host and the sequelae thereof.

## Introduction

Most people are infected with Epstein-Barr virus (EBV) during childhood or adolescence, resulting in a persistent, mostly latent EBV infection. The primary EBV infection often manifests as infectious mononucleosis (IM), especially in adolescence [1,2]. Globally EBV is causally linked to nearly 200000 incident cancers and 18000 deaths from multiple sclerosis annually [3,4], with IM elevating the risk of Hodgkin lymphoma and multiple sclerosis for unknown reasons [5–7]. Functions of EBV antibody levels as predictors of disease risk is an active field of research, see [8] and references therein.

At the same time it is unclear why upon primary EBV infection some individuals present with IM, while others do not [9]. Disease severity and duration correlate much better with e.g. CD8+ cell counts than with the viral kinetics itself and the expansion of the CD8+ cell count is controlled in asymptomatic EBV infection despite virus loads similar to those experienced in symptomatic EBV infection [1,2,10–14]. Hence current understanding suggests that IM is caused by overreaction by the immune system, rather than EBV infection per se (viremia or B-cell expansion). There now seems to be broad agreement that a massive expansion of the number of EBV-specific CD8+ cells is a characteristic of IM, while changes in the proportions of other cell populations seem less well-established [1,2,10–13,15,16]. Clinically, IM is typically characterized by fever, pharyngitis, lymphadenopathy and fatigue. The IM symptoms are believed to be caused mostly, if not entirely by the exaggerated CD8+ response [2,15,16]. Presumably IM is the same disease in teenagers as in children, because the immunological response to EBV infection is recognizably the same [10,17].

The way IM occurrence depends on age and sex is incompletely described and hard to interpret etiologically. The age distribution of incident IM is dominated by a distinct peak in the middle of the teenage years [18,19]. However, as an etiological clue this is not particularly useful because the depicted rate is not really a rate, i.e. a number of IM cases divided by the time at risk of those who have not seroconverted. Rather it is a product of the prevalence of EBV-nave persons, the hazard rate of seroconverting and the attack rate, i.e. the fraction of primary EBV infections that is accompanied by IM. Attack rates have only been estimated in young adults [20–23], and estimated sero-conversion rates are practically non-existent too.

It would therefore be valuable to devise and fit a mathematically coherent model, projecting what would be the age- and sex-specific seroconversion rate and attack rate in a hypothetical population where the observed age- and sex-specific EBV-prevalence and IM occurrence in the target population apply. Such a model could quantify e.g. how much of the IM teenage peak is due to changed behavior (changing hazard of seroconversion), and how much to

changed susceptibility to IM (changing attack rate) in teenagers compared with pre-adolescents.

As proof of concept we therefore fitted such a model based on a few large data sets, with Danes age 0–29 years in 2006–2011 as our target population.

## Methods

### Materials

We used the Danish Civil Registration System [24] to follow-up persons while resident in Denmark in 2006–2011 and of age 0–29 years for incident IM in NPR. Incident IM in the Danish National Patient Register (NPR) [25] for a person was defined as the first hospital contact with IM as main, secondary, or underlying diagnosis, classified as code 075* in ICD-8 and code B27* in ICD-10 [19,26].

In 2010 and 2011 the Danish Blood Donor Study (DBDS) [27] asked participants:"Were you ever told by a doctor that you had infectious mononucleosis" and if so–"At what age?". The study base of IM cases from DBDS was defined as DBDS participants who either 1) had reported IM in 2006–2011 at age 0–29 years or 2) had reported IM at age 0–14 years or 3) were IM cases in NPR at age 0–29 years at time of DBDS interview and born in 1976+. We then searched for these persons as IM cases in the NPR. Criterion 2) ensures that we can estimate IM hospitalization rates at age 0–17 years (donors must be 18+ years at interview) and criterion 3) makes a hospital diagnosis of IM equally valid as one proclaimed by a general practitioner.

EBV test results recorded in the Laboratory Information Management System of Statens Serum Institut, Copenhagen, Denmark were mapped into positive and negative results as in Rostgaard et al [19]. The first result was retrieved for all persons serologically tested for primary EBV infection from January 2005 to May 2011. The serological test was based on measurements of IgG antibody titers to EBV nuclear antigen (EBNA) and IgG and IgM antibody titers to EBV capsid antigen (VCA). All measurements were performed using enzyme-linked immunosorbent assay (Biotest, BioNordika, Herlev, Denmark). Each test result was coded as 1)"prior infection" if EBNA was positive, 2)"positive" if VCA IgG or VCA IgM were"positive" or"weak" while at the same time EBNA was"negative" or"weak" and 3)"negative" if VCA IgG, VCA IgM and EBNA were all"negative". Any test result that did not match any of these three disjoint criteria was discarded [19]. The test results were obtained from analyzing test samples sent from hospitals and general practitioners from all parts of Denmark, but predominantly from Sealand [19].

For convenience we used a discrete survival model and hence lumped age into 61-day intervals denoted by $a = 0,1,2,\ldots,179$, $a = 180 \approx 30$ years. The data were aggregated accordingly. In order to obey a data discretionary rule of at least 5 observations in a cell, the data were sorted by type, sex and age and cells then aggregated on a running basis to fulfil this criterion, the age interval denoted by the rounded mean of $a$. The data were used in that form and are available in **S1 Data**. We did not use data from the first year of life due to the maternally-derived EBV sero-positivity shortly after birth [17,28].

### Statistical methods

The statistical framework for this paper is a Markov model with the following states (see pages 1–25 and 457–475 in [29]):

0: EBV negative
1: EBV positive, no history of IM
2: EBV positive, a history of IM

We describe the dynamics of the system only in terms of age $a$ and sex $s$. Let $S(a,s)$ be the sex- and age-specific probability of being in state 0. Let the probability of moving from state 0 to state 1 or state 2 be $f_1(a,s)$ and $f_2(a,s)$, respectively. These are expressed in terms of the probability of being at risk in state 0, $S(a-1,s)$, the probability of moving out of state 0 if in that state, $(1+exp(-\varepsilon_s(a)))^{-1}$ and the probability of presenting with IM upon seroconversion, $P(a,s) = (1+exp(-\ddot{\imath}_s(a)))^{-1}$, i.e. $f_1(a,s) = (1-P(a,s))(1+exp(-\varepsilon_s(a)))^{-1}S(a-1,s)$ and $f_2(a,s) = P(a,s)(1+exp(-\varepsilon_s(a)))^{-1}S(a-1,s)$. Let the probability of hospitalized IM among IM cases in DBDS be $P(a,s) = (1+exp(-v_s(a)))^{-1}$. Let p0, p1 and p2 be shorthand for the probability of being in state 0, 1, and 2. Similarly let imfrac and hospfrac be shorthand for the probability of IM upon seroconversion and hospitalization upon having IM.

The model was fitted using SAS proc HPNLMOD. The functions $\varepsilon_s(a)$, and $v_s(a)$ were modeled as fractional polynomials of degree 4 and 2, with power sets (-1,0,0.5,1) and (2,1), respectively (see pages 77–98 in [30]). Thus $v_s(a)$ was a second degree polynomial in $a$. These fractional polynomials sufficed to provide an adequate fit, according to goodness-of-fit tests and inspection of residuals. Preliminary analyses revealed that $\ddot{\imath}_s(a)$ were complicated functions, requiring 8–12 degrees of freedom for an adequate fit. The functions $\ddot{\imath}_s(a)$ were modeled as restricted cubic splines (see pages 20–24 in [31]). The knots for the splines were common for the sexes and at the outset placed at deciles of the number of IM events in NPR. We then added knots at the 2.5, 5 and 7.5 percentile to obtain a satisfactory fit also in a region with few IM cases but much change in seroconversion rates. imfrac did not look as expected in the tail and very different between the sexes. We considered this to be a consequence of model uncertainty regarding the post-teenage years in combination with the notorious wigglyness of high-dimensional splines. To remedy this we therefore removed the two top knots, retaining an adequate model fit according to goodness-of-fit tests. Finally we fixed $\ddot{\imath}_s(a)$ to be constant above the new top knot, at the cost of an increase in deviance of 2.5–3 in each sex in order to remove unrealistic decreasing trends above the top knot.

The link between model and data was provided by the following contributions to the model log-likelihood (ll):

for EBV prevalence data with POS positives among N tested:

ll = POS*log(p1+p2)+(N-POS)*log(p0)

for DBDS data with POS hospitalized among N IM cases:

ll = POS*log(hospfrac)+(N-POS)*log(1-hospfrac)

for NPR data with EVENTS IM cases in PYRS person-years at risk:

ll = EVENTS*log(him)-him*PYRS

where

him = 6*imfrac/((1+exp(-))*(1+exp(-v)))*p0/(p0+p1)/0.9

The construction of most of the graphs in Fig 1 from quantities described here is immediate. The seroconversion hazard rate in Fig 1C is $6/(1+exp(-\varepsilon_s(a)))$ events per person-year.

## Assumptions

We assume that all persons start in state 0 at birth, i.e. we ignore that EBV can pass across the placenta during pregnancy [32]. Death, emigration etc is considered non-informative censoring. The incubation time of around 42 days [12] from EBV infection to possibly overt IM is ignored. Since they are few, and not directly identifiable, we have not created a special state for persons who will remain EBV-negative [33,34], e.g. due to lack of the EBV receptor CD21 on B-cells [33]. Similarly, states 1 and 2 are absorbing, so we do not allow alternation between susceptible and non-susceptible states, suggested as possible by Helminen et al. [34], nor do we allow multiple EBV infections where the first did not cause IM, but one of the later did, i.e. we

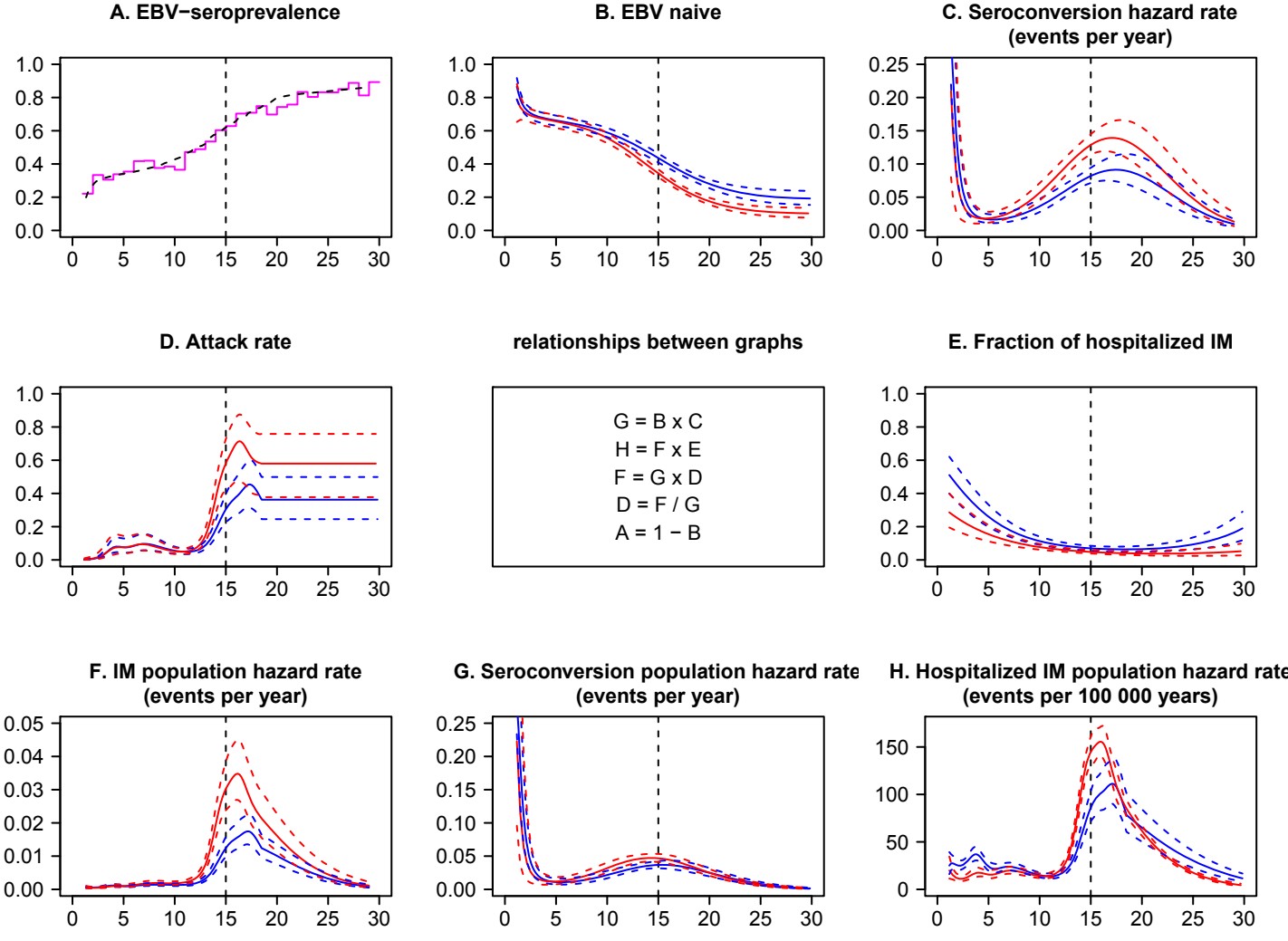

**Fig 1. Model predictions with 95% confidence limits by age for females (red) and males (blue).** The model was created from jointly fitting C, D, E; the results in B, F, G and H were derived from this. The flat attack rate above age 18 years in subgraph D is a self-imposed model constraint, see Methods. Subgraph A is the EBV-seroprevalence by age in Denmark in 2006–2011. The dotted line was predicted from the model.

assume that once a latent EBV infection is established you cannot get IM caused by EBV. We assume that a person can have IM only once, e.g. that a person cannot have a second IM caused by e.g. cytomegalovirus. The data on IM incidence will usually be exaggerated due to lack of proper laboratory confirmation of EBV involvement in IM-like disease symptoms. Part of the problem is that only 90% of true IM is caused by EBV [35], that is the 0.9 in the expression for 'him' above.

## Miscellanea

The risk of getting IM before age 30 years was calculated as *1-exp(-H(a))*, where *H(a)* is the cumulative population IM incidence rate at age *a*, i.e. the integral of the curves shown in Fig 1F.

Estimates and confidence limits as presented in the figures were calculated from the predict logic of SAS proc HPNLMOD. In these calculations the leading coefficient of $v_s(a)$ was fixed to avoid inexplicable variance inflation in Fig 1H. The variance estimates in the other graphs were essentially unaltered by this fix.

All statistical calculations were performed using SAS statistical software (SAS Institute, Cary, NC. version 9.4).

The study was approved by the institutional review board of Statens Serum Institut and the Scientific Ethics Committee Central Denmark (M2009237). As such it adheres to Danish law, including the European Union General Data Protection Regulation and is conducted according to the principles expressed in the Declaration of Helsinki. Written informed consent was obtained at enrollment into the DBDS[27], while specific informed consent for use of the other (register) data sources in this study was not needed according to Danish law.

## Results

All Danes age 0–29 years resident in Denmark somewhen during calendar years 2006–2011, in all 2,485,292 persons, were followed up in the same age and period range for a hospital contact with an IM diagnosis during 11,376,713 person-years of follow-up, yielding 4703 incidents of hospitalized IM. 2487 blood donors from The Danish Blood Donor Study, who had self-reported IM or had been hospitalized with IM under the right conditions (age and period, see Methods) were assessed for hospitalized IM to yield the fraction of hospitalized IM among IM cases (185/2487 = 7% of IM cases). 6145 persons tested for EBV antibodies at Statens Serum Institut at age 0–29 years during calendar years 2006–2011 yielded 3513 (57%) infected with EBV. The three statistically sufficient data sets for these three outcomes and the only data sets used for our analyses are available in **S1 Data**, labelled in the type variable as NPR, DBDS and EBVPREV, respectively. The results of our modeling are a set of age- and sex-specific predictions, presented in Fig 1A–1H, and the same predictions in a slightly aggregated life-table format in Table 1, with columns labelled B to H. Throughout we shall only refer to the figures, the reader may consult the relevant columns of the table instead.

The EBV prevalence in our data set was generally lower than in older unselected Danish data [36,37], but otherwise similarly distributed (Fig 1A). Sex-specific corresponding proportions of EBV nave individuals are shown in Fig 1B.

Both sexes experienced peaks in seroconversion rate as infants and as young adults (Fig 1C). The seroconversion rates for boys and girls were similar on the left side of the nadir in seroconversion rate, but girls had the highest rate to the right of the nadir (Fig 1C). The seroconversion rate peaked at age 17.2 years in females and at age 17.5 years in males.

The IM attack rate rose from practically nothing in children aged 0–2 years to represent a very common phenomenon in teenagers (Fig 1D). A peak in attack rate appeared in teenage years, and was especially pronounced among girls. The attack rate was higher in females than males throughout the teenage years. The attack rate peaked at age 16.3 years in girls and at age 17.3 years in boys and likewise the local minimum in attack rate to the left of the peak occurred at age 11.0 years in boys and at age 10.5 years in girls (Fig 1D).

For all ages the fraction of hospitalized IM cases was larger for males than for females. The fraction of IM cases becoming hospitalized was unimodal and typically low with a minimum of 6% at age 18.3 years and 4% at age 21.8 years for males and females, respectively (Fig 1E).

Fig 1F, 1G and 1H contain what we call population rates. The denominator in these rates is time at risk for the entire population, not just the subpopulation of EBV nave.

The IM population hazard rate is the product of the seroconversion population hazard rate and the attack rate. The location and shape of the IM population hazard rate peak in teenage

**Table 1. Model predictions by age and sex.** Prevalent cases per 100,000 cases (B), events per 100,000 person-years (C,F,G,H) and probability x 100,000 (D,E). B: EBV naive, C: seroconversion hazard rate, D: attack rate, E: fraction of hospitalized IM cases, F: IM population hazard rate, G: seroconversion population hazard rate and H: hospitalized IM population hazard rate. C-H are averages within the age group, B is the maximum within the age group, i.e. who were EBV-naïve at exact age 1,2,...Refer to Fig 1 and methods for further details.

| | B | | C | | D | | E | | F | | G | | H | |
|---|---|---|---|---|---|---|---|---|---|---|---|---|---|---|
| age | m | f | m | f | m | f | m | f | m | f | m | f | m | f |
| 1 | 91520 | 82765 | 24843 | 20157 | 261 | 420 | 49106 | 27478 | 52 | 60 | 20646 | 15698 | 25 | 17 |
| 2 | 73260 | 70853 | 7174 | 4502 | 1301 | 1578 | 41541 | 23364 | 63 | 49 | 5138 | 3145 | 26 | 11 |
| 3 | 69067 | 68204 | 2792 | 2208 | 4962 | 4849 | 34775 | 19871 | 100 | 78 | 1912 | 1495 | 35 | 15 |
| 4 | 67340 | 66798 | 1832 | 1802 | 7830 | 7494 | 28953 | 16939 | 107 | 100 | 1228 | 1197 | 31 | 17 |
| 5 | 66158 | 65609 | 1643 | 1903 | 7713 | 7890 | 24088 | 14497 | 93 | 109 | 1083 | 1243 | 22 | 16 |
| 6 | 65077 | 64340 | 1757 | 2287 | 9101 | 9138 | 20111 | 12473 | 115 | 149 | 1140 | 1464 | 23 | 18 |
| 7 | 63917 | 62827 | 2066 | 2917 | 9419 | 9007 | 16907 | 10802 | 138 | 182 | 1316 | 1820 | 23 | 20 |
| 8 | 62565 | 60938 | 2544 | 3795 | 8049 | 7406 | 14351 | 9423 | 141 | 188 | 1585 | 2293 | 20 | 18 |
| 9 | 60932 | 58561 | 3185 | 4921 | 6331 | 5798 | 12326 | 8287 | 135 | 183 | 1929 | 2848 | 17 | 15 |
| 10 | 58945 | 55619 | 3975 | 6272 | 5172 | 5008 | 10729 | 7351 | 133 | 191 | 2325 | 3433 | 14 | 14 |
| 11 | 56558 | 52092 | 4887 | 7786 | 4995 | 5579 | 9477 | 6581 | 152 | 247 | 2735 | 3974 | 14 | 16 |
| 12 | 53762 | 48038 | 5869 | 9368 | 6501 | 9274 | 8502 | 5947 | 226 | 455 | 3115 | 4395 | 19 | 27 |
| 13 | 50599 | 43594 | 6850 | 10893 | 12119 | 22756 | 7751 | 5426 | 462 | 1177 | 3419 | 4642 | 36 | 63 |
| 14 | 47154 | 38961 | 7747 | 12220 | 23511 | 47058 | 7186 | 5000 | 946 | 2463 | 3614 | 4712 | 68 | 123 |
| 15 | 43558 | 34368 | 8470 | 13220 | 33900 | 63392 | 6776 | 4655 | 1386 | 3240 | 3680 | 4602 | 94 | 151 |
| 16 | 39957 | 30029 | 8944 | 13790 | 40552 | 70366 | 6501 | 4378 | 1625 | 3377 | 3608 | 4318 | 106 | 148 |
| 17 | 36495 | 26105 | 9118 | 13876 | 44644 | 62818 | 6347 | 4160 | 1692 | 2714 | 3411 | 3883 | 107 | 113 |
| 18 | 33291 | 22694 | 8973 | 13474 | 38086 | 58313 | 6306 | 3994 | 1317 | 2176 | 3106 | 3357 | 83 | 87 |
| 19 | 30431 | 19826 | 8525 | 12633 | 36242 | 57974 | 6377 | 3875 | 1100 | 1810 | 2732 | 2810 | 70 | 70 |
| 20 | 27955 | 17481 | 7821 | 11446 | 36242 | 57974 | 6562 | 3800 | 939 | 1475 | 2331 | 2290 | 62 | 56 |
| 21 | 25875 | 15608 | 6932 | 10026 | 36242 | 57974 | 6871 | 3765 | 779 | 1175 | 1934 | 1824 | 53 | 44 |
| 22 | 24172 | 14143 | 5939 | 8498 | 36242 | 57974 | 7320 | 3770 | 630 | 917 | 1564 | 1423 | 46 | 35 |
| 23 | 22810 | 13017 | 4921 | 6973 | 36242 | 57974 | 7932 | 3816 | 497 | 702 | 1235 | 1089 | 39 | 27 |
| 24 | 21747 | 12166 | 3946 | 5545 | 36242 | 57974 | 8738 | 3902 | 383 | 527 | 951 | 819 | 33 | 21 |
| 25 | 20935 | 11532 | 3065 | 4276 | 36242 | 57974 | 9781 | 4033 | 288 | 389 | 716 | 604 | 28 | 16 |
| 26 | 20330 | 11070 | 2306 | 3201 | 36242 | 57974 | 11118 | 4212 | 212 | 282 | 526 | 437 | 23 | 12 |
| 27 | 19888 | 10737 | 1683 | 2327 | 36242 | 57974 | 12820 | 4444 | 152 | 200 | 377 | 310 | 19 | 9 |
| 28 | 19573 | 10503 | 1193 | 1645 | 36242 | 57974 | 14976 | 4738 | 106 | 139 | 264 | 215 | 16 | 7 |
| 29 | 19354 | 10341 | 821 | 1131 | 36242 | 57974 | 17692 | 5104 | 72 | 94 | 180 | 146 | 13 | 5 |

years (Fig 1F) was essentially determined by the attack rate (Fig 1D), which varied considerably more in this age span than the seroconversion population hazard rate (Fig 1G).

The combination of information in Fig 1C and 1D revealed several things. For children age 0–2 years the attack rate was low and the seroconversion rate high, as a priori expected from prevalence and rate data. For 3–12 years old children the IM population hazard rate was kept low mainly by a small seroconversion rate, since the attack rate, relatively speaking increased substantially compared to the attack rate in 0–2 years old children. Comparing children age 4 to 5 years (the nadir of seroconversion) with teenagers age 16 to 17 years (the peak in IM attack rate) the seroconversion rate was lower by a factor of 6 to 8, while the corresponding attack rate was lower by a factor of 6 to 10. Accordingly, the low incidence of IM in 3–12 years old children was roughly equally due to low attack rate and low seroconversion rate.

## Discussion

Our analyses for the first time provide detailed and compelling evidence that the accumulation of IM among adolescents, a characteristic of western industrialized countries, reflects age-dependent variations both in IM attack rates and EBV seroconversion hazard rates.

Both early childhood and adolescence are age-periods characterized by social behaviors involving the exchange of saliva, the primary route for EBV transmission, e.g. through sharing of toys and utensils in early childhood and through kissing in adolescence and early adulthood [38]. Deep kissing as the main route of EBV transmission in adolescence and beyond is well established [9], while the evidence for sharing of toys and utensils in early childhood as an important route of EBV transmission is weaker and indirect, e.g. a marked reduction in IM risk for each additional sibling, especially when the age-differential is small[19,39], presumably due to pre-teenage EBV infection.

To become infected, EBV nave individuals must interact with EBV-positive infectious individuals. Consequently, spreading of EBV depends on patterns of interaction between EBV-susceptible and EBV-infectious individuals and the likelihood of EBV transmission and infection at such encounters. In early childhood when the vast majority of individuals are still EBV-nave, EBV will spread rapidly because many encounters between these EBV-nave children and EBV-positive parents/adults/same age children has the potential to create an EBV infection in the EBV-nave child.

The steep decline in seroconversion hazard rates between ages 2 and 5 years is disproportionate to the decrease in EBV-susceptible individuals. Therefore, rather than the gradual reduction in proportions of susceptible and possibly also acutely infected (infectious) individuals, the decreasing risk of EBV-infection and the plateauing sero-prevalence likely reflect age-related changes in behavior associated with lower risk of EBV transmission from both other children and parents/adults.

The second wave of EBV infection occurred in adolescence through early adulthood, with the highest EBV hazard rates occurring at slightly younger ages in females than in males. This may reflect earlier puberty in girls than in boys, and the typical age-disparity in female-male relationships with girls tending to partner with older boys [40]. Because of the age-dependent increase in EBV-sero-positivity, girls at any age would tend to engage with boys more likely to be EBV infected than boys their own age, whereas the opposite would be true for boys who would engage with younger girls, less likely to be EBV infected than girls their own age. Although of waning relevance in Western societies we also speculate that girls more often than boys expose themselves as caretakers to siblings and other children of age 0–3 years, whom we have identified as risk factors for IM and thus primary EBV infection [19]. All these interaction patterns would accelerate seroconversion in the female population and decelerate it in the male population.

The shape of the attack rate essentially complies with the dogma of IM being more frequent and severe the older the age at seroconversion [41–43] yielding something close to a monotone increase by age (Fig 1D).

The mechanisms underlying the age-dependent variation in IM attack rate have remained elusive, but proposed explanations include corresponding variations in mode and dose of infection and in host immune response [2,11,15,44]. The host immune response may vary by age for at least two reasons: 1) NK cell responses may assume greater importance, and perhaps be more effective, in combating virus infections early in life [15], and/or 2) adolescents infected with EBV may recruit large numbers of cross-reactive memory T cells previously created in response to other viral infections, which may more easily be activated but be less efficient in controlling the infection than primary responses from recruited nave T cells [44].

However, in light of the very rapid change in IM attack rate, we do not consider cross-reactivity of memory T cells to be a likely major contributor to this change. Similarly Balfour et al. found no evidence of influenza-EBV dual specific CD8+ T cells in their study cohort to support this explanation [9,11]. Likewise both simulations [44] and observational studies [2,12,45] suggest that the initial viral load, and hence dose or mode of delivery is of little importance for IM risk.

Our results, including the fact that the adolescent attack rate peak among females occurs at a slightly younger age than the corresponding peak in males, instead point to IM susceptibility as somehow being subject to mechanisms that involve growth and/or sex hormones whose levels change as part of sexual maturation. In this regard, it is noteworthy that both estrogen and androgens are known to influence immune responses via epigenetic mechanisms, see [46] and references therein.

## Strengths and weaknesses

We believe the serological data on prevalent EBV status to be accurate. They are based on enzyme-linked immunosorbent assays, which can perform very similarly to the gold standard of immunofluorescence arrays [1,47–49]. However, the tested patients were not randomly sampled, and as such may yield a biased representation of the age- and sex-specific EBV-prevalence in our population. Specifically, most persons in our sample were presumably tested in order to determine whether symptoms similar to IM could be due to an acute EBV infection. Furthermore, we suspect that many of the samples were sent for serological testing due to atypical IM symptoms or results of a quick but unreliable IM test, that the general practitioner did not trust. As such one would expect to sample too many recently EBV-infected persons. On the other hand, comparison with older unselected Danish data sets suggest, if anything, that we have too few EBV-infected persons in our sample at a given age.

Secular changes, specifically the Danish society becoming more affluent would tend to lower the age-specific sero-prevalence in our material compared to older Danish materials [1,50]. This could explain the discrepancy, and recent examples of such trends in other Western countries exist [11,41]. In Denmark the gradual increase in childcare attendance from around 1965 to 2000 [51] would tend to work in the opposite direction, but the effect is probably modest since the most common type of childcare for children age 0–2 years is by daycare mothers, i.e. caretakers taking care of only a few children. Currently only a third of a generation of children age 0–2 years attends an institution (creche, kindergarten or integrated institution), see {**http://statistikbanken.dk**}{**statistikbanken.dk**}.

Altogether, we believe that our estimated seroconversion rates are sufficiently accurate to model the essential seroconversion dynamics in our target population.

For the purpose of attributing causes for IM in different age groups, it seems more important to get the ratios of age-specific attack rates within sexes, rather than the exact level, correct. We see no reason why our data or modeling should be noticeably biased with respect to assessing ratios of age-specific attack rates within sexes. Furthermore, our estimated attack rates around age 20 years are compatible with earlier detailed longitudinal studies on university students and army recruits [20–22], and do not de facto become 100% at any age as would be the sign of a severe upwardly biased ascertainment of incident IM.

We believe the variation in the fraction of IM cases hospitalized to be a natural screening phenomenon. Specifically, we believe that general practitioners expect IM symptoms in teenagers to be caused by IM and therefore do not admit such cases to hospital, while the more unexpected and for children more non-specific IM symptoms [52–55] would cause general practitioners to admit a patient to hospital for further investigation more often. We do not

know why the fraction of hospitalized IM cases is higher in boys than in girls; if anything, girls seem on average to have the most vigorous immune response as measured by EBV antibody titers [56–58]. Furthermore, there seemingly is no age gradient (age 6–17 years) in EBV antibody titers [58], supporting the view that the bathtub shaped curves are a screening phenomenon, rather than due to physiology.

The cumulative risk of IM before age 30 years was 13.3% for males and 22.4% for females. This estimate is quite high compared to other estimates ($\approx$ 5% with much variation) (Rostgaard et al. [26] and Table 5 in Hjalgrim [59]). We have no immediate explanation for this. However, we do not consider it surprising to have a substantially larger "lifetime" risk of IM in our target population than in other older and less affluent settings referred to above. E.g. the percentage of 15–17 year old EBV nave Americans increased from 22 to 31 over just 6 years (Table 2 in Balfour et al. [1]), which all else equal should increase the occurrence of IM in that age span a factor 31/22 = 1.41. If the percentage of EBV nave at the IM teenage peak was much lower in the past a change in the occurrence of IM of a factor 3 or 4 is certainly possible. Furthermore the blood donors in our study being on average better educated and wealthier than non-donors [60] would suggest them to be recruited from affluent population strata and as such more prone to late EBV infection and thus presenting with IM than the general population.

## Conclusion

Studies to predict the possible benefit of a specific EBV vaccine was one of five priorities outlined at an EBV-vaccine meeting organized by the US National Institutes of Health in 2011 [61]. The present study provides for the first time some of the knowledge needed for that purpose by precisely displaying at what age persons seroconvert and when it has consequences in terms of IM, with all the sequelae that goes with that [3–7].

Mathematically the pair of descriptors (EBV hazard rate, IM attack rate) has the advantage compared with (EBV prevalence, IM incidence) of being more"local" in time, and therefore better suited to generation of causal interpretations and hypotheses, as causal mechanisms work locally in time, i.e. causes continually transmit their effects [62]. We think our study vindicates this point of view.

Methodologically we found it relatively easy to transform prevalence data into mathematically coherent and equivalent forms, primarily smooth hazard functions. We found these more informative than the raw prevalence data for, in this case, the dynamics of EBV infection. Our prevalence data were very detailed regarding age, but usually much cruder data would suffice for obtaining a model-based smooth hazard function. We believe that this type of analysis would be helpful in many future studies of the epidemiology of specific persistent infections.

## Supporting information

**S1 Data. The raw data for model fitting.**
(TXT)

## Author Contributions

**Conceptualization:** Klaus Rostgaard.

**Data curation:** Christian Erikstrup, Ole Pedersen, Henrik Ullum, Lars Peter Nielsen, Marianne Voldstedlund, Henrik Hjalgrim.

**Formal analysis:** Klaus Rostgaard.

**Funding acquisition:** Klaus Rostgaard, Henry H. Balfour, Jr., Henrik Hjalgrim.

**Investigation:** Klaus Rostgaard, Henry H. Balfour, Jr., Ruth Jarrett, Henrik Hjalgrim.

**Methodology:** Klaus Rostgaard.

**Resources:** Christian Erikstrup, Ole Pedersen, Henrik Ullum, Lars Peter Nielsen, Marianne Voldstedlund, Henrik Hjalgrim.

**Software:** Klaus Rostgaard.

**Visualization:** Klaus Rostgaard.

**Writing – original draft:** Klaus Rostgaard, Henry H. Balfour, Jr., Ruth Jarrett, Henrik Hjalgrim.

**Writing – review & editing:** Klaus Rostgaard, Henry H. Balfour, Jr., Ruth Jarrett, Christian Erikstrup, Ole Pedersen, Henrik Ullum, Lars Peter Nielsen, Marianne Voldstedlund, Henrik Hjalgrim.

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
