## [Decision Letter · Decision Letter 0]

4 Oct 2019

PONE-D-19-25327

Primary Epstein-Barr virus infection with and without infectious mononucleosis

PLOS ONE

Dear Mr. Rostgaard,

Thank you for submitting your manuscript to PLOS ONE. After careful consideration, we feel that it has merit but does not fully meet PLOS ONE’s publication criteria as it currently stands. Therefore, we invite you to submit a revised version of the manuscript that addresses the points raised during the review process.

We would appreciate receiving your revised manuscript by Nov 18 2019 11:59PM. To enhance the reproducibility of your results, we recommend that if applicable you deposit your laboratory protocols in protocols.io, where a protocol can be assigned its own identifier (DOI) such that it can be cited independently in the future. For instructions see: http://journals.plos.org/plosone/s/submission-guidelines#loc-laboratory-protocols

We look forward to receiving your revised manuscript.

Kind regards,

Gulfaraz Khan, PhD, FRCPath

Academic Editor

PLOS ONE

Journal Requirements:

2. Our internal editors have looked over your manuscript and determined that it may be within the scope of our Mathematical Modelling of Infectious Disease Dynamics Call for Papers. The Collection will encompass a diverse range of research articles on using mathematical models to better understand infectious diseases. Additional information can be found on our announcement page: https://collections.plos.org/s/mathematical-disease-dynamics. If you would like your manuscript to be considered for this collection, please let us know in your cover letter and we will ensure that your paper is treated as if you were responding to this call. If you would prefer to remove your manuscript from collection consideration, please specify this in the cover letter.

3. Please note that PLOS ONE has specific guidelines on software sharing (http://journals.plos.org/plosone/s/materials-and-software-sharing#loc-sharing-software) for manuscripts whose main purpose is the description of a new software or software package. In this case, new software must conform to the Open Source Definition (https://opensource.org/docs/osd) and be deposited in an open software archive. Please see http://journals.plos.org/plosone/s/materials-and-software-sharing#loc-depositing-software for more information on depositing your software.

Additional Editor Comments (if provided):

Reviewers' comments:

Reviewer's Responses to Questions

**Comments to the Author**

1. Is the manuscript technically sound, and do the data support the conclusions?

Reviewer #1: Partly

Reviewer #2: Yes

2. Has the statistical analysis been performed appropriately and rigorously? 

Reviewer #1: I Don't Know

Reviewer #2: Yes

3. Have the authors made all data underlying the findings in their manuscript fully available?

Reviewer #1: Yes

Reviewer #2: Yes

4. Is the manuscript presented in an intelligible fashion and written in standard English?

Reviewer #1: No

Reviewer #2: Yes

5. Review Comments to the Author

Reviewer #1: It would be useful to report the number of individuals included in the study, and a descriptive table showing the number of individuals who are EBV positive or negative at each age. A multiple-decrement life table could provide at a glance a good sense of the infection and disease process, and thus a more convincing presentation of the results presented.

Reviewer #2: Manuscript Nr.: PONE-D-19-25327

Rostgaard et al., "Primary Epstein-Barr virus infection with and without infectious mononucleosis"

The authors evaluate the age and sex dependency of infectious mononucleosis (IM) in the Danish population between 2006 and 2011 for the ages between 0 and 29 years. IM is identified by hospitalization or self-reporting. The Epstein Barr virus (EBV) serostatus is assessed by IgG and IgM responses to virus capsid antigen (VCA) and IgG to the nuclear antigen 1 of EBV (EBNA1). They report that the cumulative IM risk for males before 30 years of age is 13% and for females 22%. IM peaks during puberty at 16.3 years earlier in girls than with 17.3 years in boys. The authors suggest that alterations in the immune system during puberty might account for the increased risk to develop the immunopathology of IM in response to primary EBV infection.

The reported data and the differences between girls and boys are interesting. One wonders if however quantitative data could be obtained from the assessed serotests and if other virus specific serologies were reported in at least a subgroup of individuals.

Major comments:

1. As pointed out by the authors overall viral loads do not differ between individuals with asymptomatic or symptomatic EBV infection. However, the composition of the primary infection might be altered during IM. Do the authors find any evidence that seroconversion during adolescence irrespective of IM results in an altered ratio of VCA (lytic infection) versus EBNA1 (latent and lytic infection) specific antibody titers?

2. Additional infections have been proposed to mature or alter the human immune system. Was the serostatus to other pathogens assessed and shows a similar bimodal distribution with two peaks of acquisition (0-2 years and during puberty)?

3. The increased rate of IM related hospitalization in males compared to females despite higher IM incidence in females is puzzling and could point towards more severe immunopathology in males during IM. Do the authors observe higher immunoglobulin titers against EBV antigens in males compared to females?

Minor comments:

1. In their results the authors often stay vague by just stating that one incidence rate is higher than another etc. I think it would be helpful if the manuscript would indicate the actual numbers that the deployed modelling generated.

In summary, this is an interesting manuscript on IM in the Danish population, reporting a surprisingly high cumulative risk in females with a fairly narrow IM peak around puberty. The study could be further improved by reporting some of the available immune parameters quantitatively.

6. PLOS authors have the option to publish the peer review history of their article (what does this mean?). If published, this will include your full peer review and any attached files.

Reviewer #1: No

Reviewer #2: No

---

## [Author Response · Author response to Decision Letter 0]

11 Nov 2019

We thank the editor and the reviewers for this opportunity to make the presentation in our manuscript more informative. We hope our solutions will satisfy all parties and lead to the acceptance of the revised manuscript for publication.

5. Review Comments to the Author

Reviewer #1: It would be useful to report the number of individuals included in the study, and a descriptive table showing the number of individuals who are EBV positive or negative at each age. A multiple-decrement life table could provide at a glance a good sense of the infection and disease process, and thus a more convincing presentation of the results presented.

A: We have created a life-table with cells by sex and 1-year age group, containing estimates on the form N per 100000 as per life table tradition of all the model predictions in the figure. The new table and the original figures thus form a complementary presentation of our findings. We have added a new first paragraph to the results section, introducing the figures and the new life-table and providing totals of persons contributing to various parts of the model. The new paragraph reads:

“All Danes age 0-29 years resident in Denmark somewhen during calendar years 2006-2011, in all 2,485,292 persons, were followed up in the same age and period range for a hospital contact with an IM diagnosis during 11,376,713 person-years of follow-up, yielding 4703 incidents of hospitalized IM. 2487 blood donors from The Danish Blood Donor Study, who had self-reported IM or had been hospitalized with IM under the right conditions (age and period, see Methods) were assessed for hospitalized IM to yield the fraction of hospitalized IM among IM cases (185/2487=7% of IM cases). 6145 persons tested for EBV antibodies at Statens Serum Institut at age 0-29 years during calendar years 2006-2011 yielded 3513 (57%) infected with EBV. The three statistically sufficient data sets for these three outcomes and the only data sets used for our analyses are available in S1 Data, labelled in the type variable as NPR, DBDS and EBVPREV, respectively. The results of our modeling are a set of age- and sex-specific predictions, presented in Fig 1A-1H, and the same predictions in a slightly aggregated life-table format in Table 1, with columns labelled B to H. Throughout we shall only refer to the figures, the reader may consult the relevant columns of the table instead.” 

Reviewer #2: Manuscript Nr.: PONE-D-19-25327

Rostgaard et al., "Primary Epstein-Barr virus infection with and without infectious mononucleosis"

The authors evaluate the age and sex dependency of infectious mononucleosis (IM) in the Danish population between 2006 and 2011 for the ages between 0 and 29 years. IM is identified by hospitalization or self-reporting. The Epstein Barr virus (EBV) serostatus is assessed by IgG and IgM responses to virus capsid antigen (VCA) and IgG to the nuclear antigen 1 of EBV (EBNA1). They report that the cumulative IM risk for males before 30 years of age is 13% and for females 22%. IM peaks during puberty at 16.3 years earlier in girls than with 17.3 years in boys. The authors suggest that alterations in the immune system during puberty might account for the increased risk to develop the immunopathology of IM in response to primary EBV infection.

The reported data and the differences between girls and boys are interesting. One wonders if however quantitative data could be obtained from the assessed serotests and if other virus specific serologies were reported in at least a subgroup of individuals.

A: Our data material was collected with the sole purpose of illuminating IM epidemiology. We therefore do not have access to other microbiological test result regarding (a subset of) our EBV tested cohort. Furthermore, all our test results regarding the specific biomarkers are of a semi-qualitative nature: “positive”, “weakly positive”, “negative” etc. We therefore are unable to address the following reviewer questions based on our own biomarker materials. We have instead tried to address the questions based on the available literature, and added content to the discussion when possible and informative.

1. As pointed out by the authors overall viral loads do not differ between individuals with asymptomatic or symptomatic EBV infection. However, the composition of the primary infection might be altered during IM. Do the authors find any evidence that seroconversion during adolescence irrespective of IM results in an altered ratio of VCA (lytic infection) versus EBNA1 (latent and lytic infection) specific antibody titers?

A: We have not been able to find relevant references on the specific question. And unfortunately, as far as we can see, we do not have data available that would at least put us in a position to suggest an answer to it. We have therefore decided only to flag this interesting avenue of research. We have added to the end of the first paragraph of the introduction:

“Functions of EBV antibody levels as predictors of disease risk is an active field of research, see REF and references therein.” 

2. Additional infections have been proposed to mature or alter the human immune system. Was the serostatus to other pathogens assessed and shows a similar bimodal distribution with two peaks of acquisition (0-2 years and during puberty)?

A: As we do not have such samples, and since we consider the evidence to be had from other infections for an age-dependently altered immune system as part explanation for the bimodal EBV seroconversion rate as weak we have not changed the text.

3. The increased rate of IM related hospitalization in males compared to females despite higher IM incidence in females is puzzling and could point towards more severe immunopathology in males during IM. Do the authors observe higher immunoglobulin titers against EBV antigens in males compared to females?

A: To the end of the fifth paragraph in “Strengths and weaknesses” we have added:

“We do not know why the fraction of hospitalized IM cases is higher in boys than in girls; if anything, girls seem on average to have the most vigorous immune response as measured by EBV antibody titers (REFS). Furthermore, there seemingly is no age gradient (age 6-17 years) in EBV antibody titers (REF), supporting the view that the bathtub shaped curves are a screening phenomenon, rather than due to physiology.”

Minor comments:

1. In their results the authors often stay vague by just stating that one incidence rate is higher than another etc. I think it would be helpful if the manuscript would indicate the actual numbers that the deployed modelling generated.

A: Done. See answer to reviewer 1.

In summary, this is an interesting manuscript on IM in the Danish population, reporting a surprisingly high cumulative risk in females with a fairly narrow IM peak around puberty. The study could be further improved by reporting some of the available immune parameters quantitatively.

---

## [Decision Letter · Decision Letter 1]

27 Nov 2019

Primary Epstein-Barr virus infection with and without infectious mononucleosis

PONE-D-19-25327R1

Dear Dr. Rostgaard,

We are pleased to inform you that your manuscript has been judged scientifically suitable for publication and will be formally accepted for publication once it complies with all outstanding technical requirements.

With kind regards,

Gulfaraz Khan, PhD, FRCPath

Academic Editor

PLOS ONE

Additional Editor Comments (optional):

Reviewers' comments:

Reviewer's Responses to Questions

**Comments to the Author**

1. If the authors have adequately addressed your comments raised in a previous round of review and you feel that this manuscript is now acceptable for publication, you may indicate that here to bypass the “Comments to the Author” section, enter your conflict of interest statement in the “Confidential to Editor” section, and submit your "Accept" recommendation.

Reviewer #1: All comments have been addressed

Reviewer #2: All comments have been addressed

2. Is the manuscript technically sound, and do the data support the conclusions?

Reviewer #1: (No Response)

Reviewer #2: Yes

3. Has the statistical analysis been performed appropriately and rigorously? 

Reviewer #1: (No Response)

Reviewer #2: Yes

4. Have the authors made all data underlying the findings in their manuscript fully available?

Reviewer #1: (No Response)

Reviewer #2: Yes

5. Is the manuscript presented in an intelligible fashion and written in standard English?

Reviewer #1: (No Response)

Reviewer #2: Yes

6. Review Comments to the Author

Reviewer #1: (No Response)

Reviewer #2: Manuscript Nr.: PONE-D-19-25327R1

Rostgaard et al., "Primary Epstein-Barr virus infection with and without infectious mononucleosis"

The authors evaluate the age and sex dependency of infectious mononucleosis (IM) in the Danish population between 2006 and 2011 for the ages between 0 and 29 years. IM is identified by hospitalization or self-reporting. The Epstein Barr virus (EBV) serostatus is assessed by IgG and IgM responses to virus capsid antigen (VCA) and IgG to the nuclear antigen 1 of EBV (EBNA1). They report that the cumulative IM risk for males before 30 years of age is 13% and for females 22%. IM peaks during puberty at 16.3 years earlier in girls than with 17.3 years in boys. The authors suggest that alterations in the immune system during puberty might account for the increased risk to develop the immunopathology of IM in response to primary EBV infection.

The revised manuscript version continues to report an interesting dichotomy between girls and boys for IM with a higher frequency and earlier onset in girls, but more frequent hospitalization of boys. Even so the authors could not provide quantitative data on the measured serologies, they have discussed these outstanding issues and now report their data more quantitatively. Therefore, the revised manuscript is improved.

Minor comment:

1. The title could be more informative and reflect some findings of the manuscript. Something along the lines of “Earlier onset and higher frequency of infectious mononucleosis in Danish females than males” would seem appropriate.

7. PLOS authors have the option to publish the peer review history of their article (what does this mean?). If published, this will include your full peer review and any attached files.

Reviewer #1: No

Reviewer #2: Yes: Christian Münz

---

## [Editor Report · Acceptance letter]

9 Dec 2019

PONE-D-19-25327R1 

Primary Epstein-Barr virus infection with and without infectious mononucleosis 

Dear Dr. Rostgaard:

I am pleased to inform you that your manuscript has been deemed suitable for publication in PLOS ONE. Congratulations! Your manuscript is now with our production department. 

With kind regards,

on behalf of

Prof Gulfaraz Khan 

Academic Editor

PLOS ONE